# Physical Modeling of Spatial and Temporal Development of Local Scour at the Downstream of Bed Protection for Low Froude Number

**Sung Won Park [1], Jin Hwan Hwang [2]** and **Jungkyu Ahn [1,***

[1] Department of Civil and Environmental Engineering, Incheon National University, 119 Academy-ro, Yeonsu-gu, Incheon 22012, Korea; billy1006@gmail.com

[2] Department of Civil and Environmental Engineering, Seoul National University, 1 Gwanak-ro, Gwanak-gu, Seoul 08826, Korea; jinhwang@snu.ac.kr

* Correspondence: ahnjk@inu.ac.kr; Tel.: +82-32-835-8084

**Abstract:** Local scour at the downstream of the river bed protection is one of the most important parameters for the design criteria and sustainable management of the hydraulic structures. Previously, various researches on its process in the equilibrium state have been suggested with experimental and numerical approaches. In this study, relatively long-term laboratory experiments of local scouring at the downstream of fixed bed in an open channel were conducted with mono-granular sediment bed and analyzed about maximum scour depth and its temporal development. In particular, we conducted experiments with relatively low Froude number (less than 0.5) and their duration of tests was exceeded over 700 hours. We modified the relationship between the dimensionless time and length scales of the maximum scour depth of the local scour hole based on the turbulent shear layer thickness. A new functional relationship between dominant factors and the maximum scour depth in the equilibrium state were suggested and compared with previously suggested formula. Also, from the results by nonlinear regression, Froude number was founded as a dominant factor on the prediction of equilibrium maximum scour depth.

**Keywords:** local scour; equilibrium state; maximum scour depth; laboratory experiment; non-linear regression analysis; Froude number

## 1. Introduction

Local scour at the downstream of the river bed protection is one of the most significant phenomena which should be considered and predicted reasonably to design and manage the upstream hydraulic structures, such as sill, weir, and drop structures in alluvial streams. Especially, undesirable channel bed scour around the structures can threaten the safety of hydraulic structures. Therefore, the total process of local scour has a great effect on hydraulic engineering problems. Local scour at the downstream of the bed protection is commonly and directly occurs by the impact of the hydraulic structure and/or its operation. Accordingly, there are several dimensions which can be representative for the scale of local scour at the downstream of the bed protection. The longitudinal length of scour hole (e.g., longitudinal length from the longitudinal transition to the first initial bed elevation at each time step) and the maximum scour depth (e.g., the first minimum value of bed elevation from the longitudinal transition at each time step) are the most important scales of local scour based on the excessive loss of bed materials. And the bed slope from the longitudinal transition between bed protection and movable bed to the longitudinal distance which has the first deepest point, upstream scour slope, and it affects the stability of the upstream structures. Also, it was revealed to have reached an equilibrium state faster than other scour hole geometric characteristics [1]. Meanwhile, the eroded bed materials are

transported downstream and deposited. This process can affect the dimensions of the scour hole. Especially, the maximum scour depth is the most important dimension for the design of upstream hydraulic structures such as bed protection and has been considered dominantly in previous researches either or both numerical and experimental approaches [2–11]. Previously, the scour depth was analyzed with consideration of the characteristic length and time scales as follows [2,6,7,12].

$$\frac{y_m(t)}{\lambda} = \left(\frac{t}{t_\lambda}\right)^\gamma \tag{1}$$

where, $y_m$ are the longitudinal length of scour hole, $t$ is time and $\lambda$ is characteristic length scale. $t_\lambda$ is characteristic time at which $y_m = \lambda$. $\gamma$ is a parameter which was suggested empirically as 0.27–0.35 by Mosonyi and Schoppmann [13], 0.34–0.40 by Dietz [3], 0.4–0.8 by van der Meulen and Vinjé [4], 0.2–0.4 by Hoffmans and Verheij [6]. However, one of the limitations of using Equation (1) is that it can be only applicable in the development phase of the temporal evolution of maximum scour depth [12] and the scour process is drastically dependent on values of $\gamma$. Also, they assumed that $\lambda$ equals the largest eddy size, which is the same as water depth at the end of bed protection. Because the range of $y_m$ in their approaches was from 2.5 to 5.0 times of water depth, the assumption about $\lambda$ in Equation (1) is not appropriate to predict the local scour development. It is necessary to analyze the local scour in the state which flow velocity is relatively slow with gate operation. Therefore, in this study, we focused on the case, which the water depth is quite deeper than expected $y_m$ due to the water control structures at the far downstream with low Froude number.

For the reasonable design of the bed protection as mentioned previously, the equilibrium value of $y_m$ is also considered dominantly. Previously Dietz [3] suggested the semi-empirical equation of equilibrium-maximum scour depth from the continuity equation as follows:

$$\frac{y_{m,e}}{h_0} = \frac{\omega U_0 - U_c}{U_c} \tag{2}$$

where, $y_{m,e}$ is equilibrium-maximum scour depth, $\omega$ is the turbulence coefficient, $U_0$ and $U_c$ are mean flow velocity at the longitudinal transition and the critical shear velocity from the Shields' diagram, respectively. The maximum scour depth increases with time and eventually reaches to so called equilibrium state when the scour depth remains as a constant value of $y_{m,e}$. Dietz [3] reported that the maximum value of the turbulence coefficient $\omega$ was amounted to $\omega = 1 + 3r_0$ and measurements showed that the average value of $\omega$ was about to $\omega = 2/3 + 2r_0$. Also following Popova [14], $\omega$ is given by $0.87 + 3.25\text{Fr} + 0.3r_0$ (Fr and $r_0$ are Froude number and depth-integrated relative turbulence intensity, $\sqrt{k_0/U_0}$, respectively). Among empirical functions, $r_0$ at the longitudinal transition was introduced as one of the most important factors for the sediment transport in the previous researches [2,3,6,14] and it can be estimated from the depth-averaged value of turbulent kinetic energy as follows:

$$k_0 = \frac{1}{h_0}\int_0^{h_0} k(z)dz \tag{3}$$

where, $z$ is vertical distance from the bottom. $k$ is the turbulence kinetic energy per mass (m$^2$/sec$^2$) which can be calculated from turbulence intensities of the 3-D flow velocity data at each point and defined as $k = 0.5\left(\overline{u'^2} + \overline{v'^2} + \overline{w'^2}\right)$ with $u'$, $v'$, and $w'$ are instantaneous flow velocity fluctuations from time-averaged one in longitudinal, lateral, and vertical directions, respectively. 3-D flow velocity measurements in the scoured hole were conducted when the maximum value of scour rate was less than $d_{50}$ per 12 hours.

Additionally, the other empirical equation of normalized $y_{m,e}$ was previously suggested with regression analysis on experimental data by Gaudio et al. [15] as follows:

$$\frac{y_{m,e}}{H_s} = 0.189\left(\frac{a_1}{\Delta d_{50}}\right) + 0.266 \tag{4}$$

where, $H_s$ is the critical specific energy in proximity of the sill $\left(= 1.5\sqrt[3]{q^2/g}\right)$, $q$ is the discharge per unit width, $g$ is gravitational acceleration, $H_s$ is the morphological jump $(= (S_1 - S_0)L)$ and the range of $a_1/H_s$ is from 1.3 to 9.1, $S_1$ and $S_0$ are, respectively, the initial and equilibrium longitudinal bed slope. $L$ is the distance between two successive sills, $\Delta = \rho_s/\rho_w$ is the relative submerged particle density, $\rho_s$ and $\rho_w$ are, respectively, the submerged sediment and water density. $d_{50}$ denotes the grain size for which 50% of the total weight of the sediment is finer. Gaudio and Marion [16] insisted that the Froude number has no influence on the dimensions of the scour hole and its temporal development. However, the range of the Froude number, which they analyzed, is quite large, 0.64–0.93. Therefore, the local scour with flow conditions of relatively low Froude number (less than 0.5) due to various operations of hydraulic structures should be analyzed to reveal the influence on the local scour. Lenzi et al. [8] analyzed several experiments about the presence of bed sills upon the scouring process and profiles. Also, they revealed that the length of the scour hole may be the dominant scale of the whole process until the equilibrium condition was reached, which is controversial.

## 2. Local Scouring and Its Process

A typical process and its notations which should be dominantly considered were schematized in Figure 1.

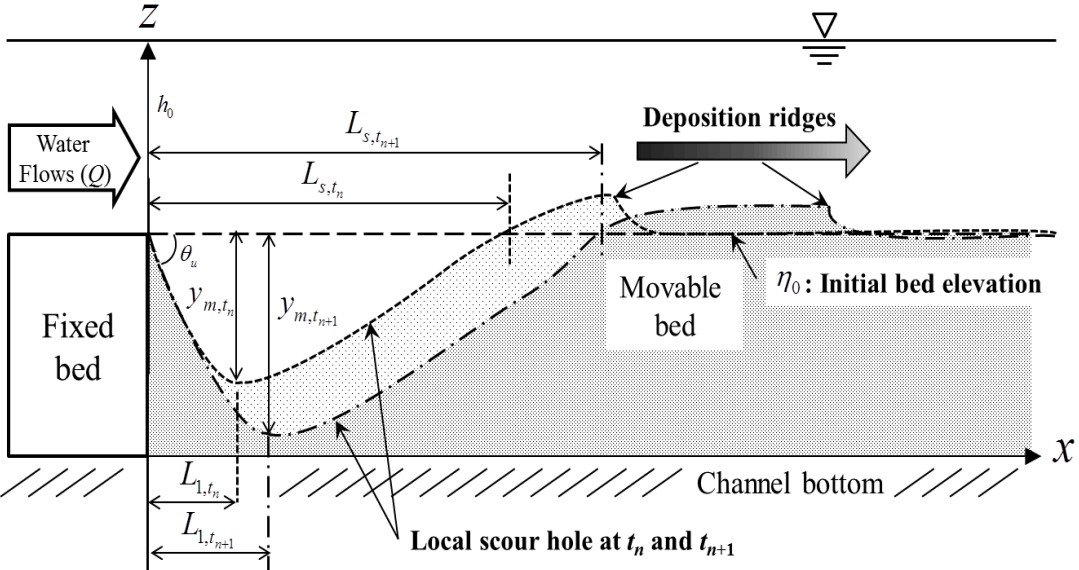

**Figure 1.** Schematic diagram of local scouring at the downstream of fixed bed (side view). $Q$ is inlet discharge from the upstream of model and $h_0$ is water depth at the longitudinal transition. $L_s$ is the longitudinal length of scour hole and $L_1$ is the longitudinal distance from the transition ($x = 0$) to the longitudinal location of $y_m$ at each time step. $\theta_u$ denotes the upstream scour depth and $\eta_0$ denotes initial bed elevation of the model.

Previous clear-water scour experiments (no sediment supply from the upstream) by Breusers [2] and Dietz [3] suggested four phases of the temporal evolution of maximum scour depth (Figure 2): initial phase, development phase, stabilization phase and equilibrium phase. In this study, physical test of local scouring with no sediment supply was conducted and analyzed the changes of local scour

hole temporally and spatially to find a functional relationship between dominant parameters and maximum scour depth.

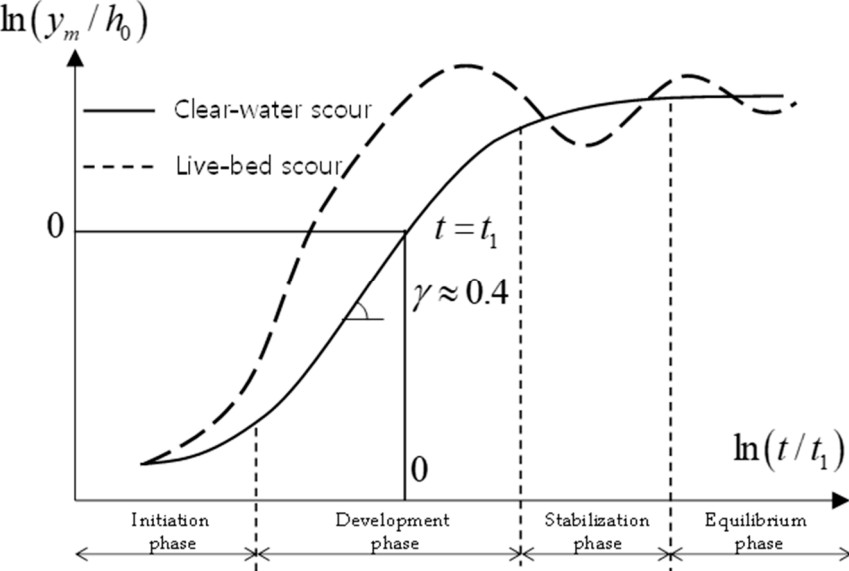

**Figure 2.** Temporal phases of normalized maximum scour depth [6]. $\gamma$ is an exponent of the relationship between dimensionless maximum scour depth and time and also the scouring rate.

## 3. Materials and Methods

### 3.1. Laboratory Flume and Measurement Devices

Laboratory experiments were carried out in a recirculating, 17.5 m long, 0.6 m wide, and 0.8 m deep flume. A schematic diagram of the flume and its system is shown in Figure 3. A head tank for stabilization of water supply, 2.9 m long, 1.2 m wide, and 1.5 m high, was connected to the flume inlet and two perforated panels and submerged wall, which can effectively remove irregular water surface undulation induced by the circulating pipe. A pair of sump tanks for water supply, storage, and sediment entrapment was installed below the flume bottom and they were connected to a closed conduit. Length of open channel for the test was 15.0 m. At the end of the flume, a tailgate which was made of a steel plate was setup to control water surface elevation. To conduct the experiments, approaching wedge model for the stabilization of inflow was positioned at 3.0 m behind the head tank of the flume and 4.0 m long and 0.2 m high acrylic box model for the protection of initial erosion was attached behind the wedge. 8.5 m long test section, which was installed at 2.5 m upstream from the tailgate of the flume, was filled with sand. Discharge from the inlet was monitored by ultrasonic flow-meter in the middle of inlet pipe and controlled by control panel automatically (Figure 3; [17]).

Carriage on the both side rails of the channel walls was setup for moving and mounting the experimental apparatus. On the carriage, the two-axis auto-traverse system for positioning span-wise and water-depth direction was mounted. The auto-traverse system can be controlled with the digital controller, which is connected to a personal computer and the carriage was moved and fixed manually. The auto-traverse has the riveting plate for the positioning measuring apparatus. Temporal change of bed elevation and three-dimensional velocity profiles at the longitudinal transition were measured by Vectrino (Nortek™, Vangkroken, Rud, Norway) on the auto-traverse system. Distance from the transmitter of the Vectrino to the eroded bed was measured at each longitudinal measuring grid and these measured values were converted into the scoured depth with respect to the longitudinal position (Figure 4).

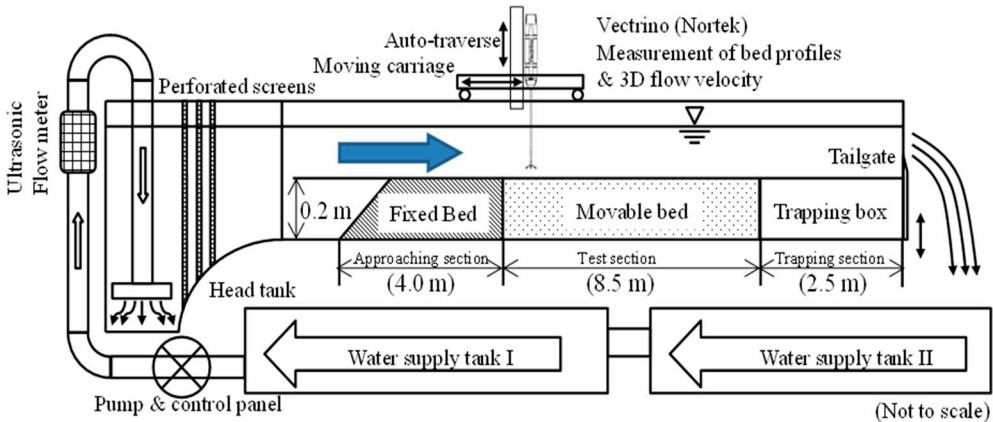

**Figure 3.** Schematic diagram of experimental channel and apparatus (side view).

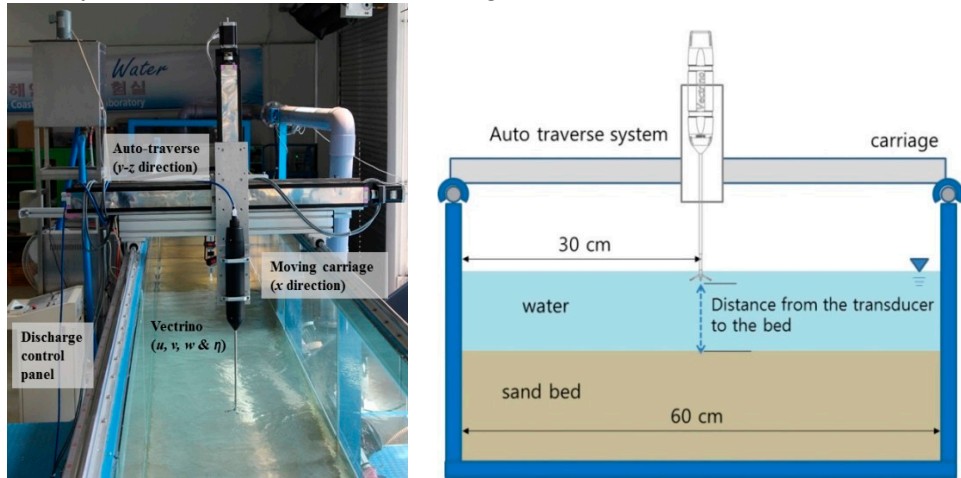

**Figure 4.** Arrangement of 3-D velocity and bed elevation measurement (left: sensor arrangement; right: schematic diagram).

### 3.2. Test Procedures and Hydraulic Conditions

At the beginning of the scour test, appropriate filling of sand bed in the test section is the most important procedure. Firstly, dry sand in the test section was filled, packed, and flattened into initial bed elevation ($\eta$ = 0.2 m) from the channel bottom and then the desirable value of water depth was reached with tap water supply to keep the initial movable bed elevation. Secondly, elevation of movable bed was flattened again to reduce the soil pore with slight compaction. Then applying small water discharges in which no significant sediment load in each case occurred were supplied gradually. Lastly main scour tests were started (Figure 5).

The experimental conditions and hydraulic parameters were described in Table 1. Ranges of water discharge at the upstream and water depth at the downstream of the flume were 0.020–0.035 and 0.12–0.15, respectively. $Q$ and $h_0$ represented the inlet discharge at the inlet pipe and water depth (m) at the end of the acrylic fixed bed ($x$ = 0) respectively. $U_0$ is cross-sectional velocity at the transition (= $Q/A$). $A$ is the cross-sectional area and $w$ is channel width. Fr is the Froude number (= $U_0 / \sqrt{gR_h}$), Re denotes the Reynolds number (= $U_0 R_h / v$) and $v$ is kinematic viscosity (= $\mu / \rho_w$, $10^{-6}$ m²/s in 20 °C water). $\mu$ is the dynamic viscosity. As mentioned previously, because a range of the Froude number in this study is relatively lower than one of other previous studies, it took much longer time to perform the test cases than one of the previous experimental studies.

| a) compacting | b) flattening | c) water supply | d) measuring |
|---|---|---|---|

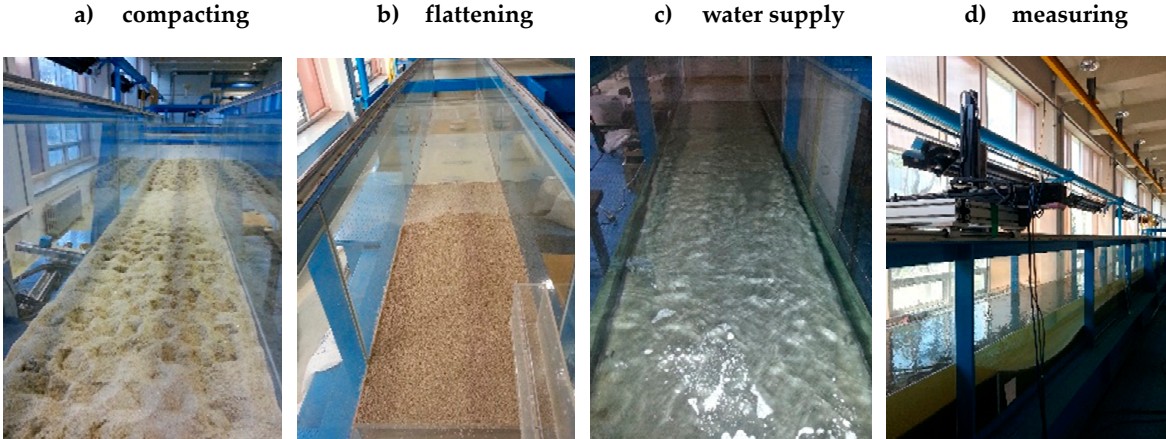

**Figure 5.** Preparation of scour test with movable bed and water supply.

**Table 1.** Hydraulic conditions of test.

| Case Number | $Q$ (m³/s) | $h_0$ (m) | $d_{50}$ (mm) | $U_0$ (m/s) | Fr (-) | $R_h$ (m) | Re ($\times 10^5$) |
|---|---|---|---|---|---|---|---|
| Q20h120d12 | 0.020 | 0.120 | | 0.278 | 0.256 | 0.086 | 0.333 |
| Q30h150d12 | 0.030 | 0.150 | | 0.333 | 0.275 | 0.100 | 0.500 |
| Q30h120d12 | 0.030 | 0.120 | 1.2 | 0.417 | 0.384 | 0.086 | 0.500 |
| Q35h120d12 | 0.035 | 0.120 | | 0.486 | 0.448 | 0.086 | 0.583 |
| Q35h144d12 | 0.035 | 0.144 | | 0.405 | 0.341 | 0.097 | 0.583 |
| Q30h145d12 | 0.030 | 0.145 | | 0.345 | 0.289 | 0.098 | 0.500 |
| Q20h120d06 | 0.020 | 0.120 | | 0.278 | 0.256 | 0.086 | 0.333 |
| Q35h120d06 | 0.035 | 0.120 | 0.6 | 0.486 | 0.448 | 0.086 | 0.583 |
| Q35h150d06 | 0.035 | 0.150 | | 0.389 | 0.321 | 0.100 | 0.583 |
| Q32h120d06 | 0.032 | 0.120 | | 0.445 | 0.410 | 0.086 | 0.533 |

## 4. Experimental Results

### 4.1. Flow and Sediment Characteristics

To analyze the functional relationships between hydraulic parameters and local scour, appropriate non-dimensional factors should be suggested. In Figure 6, non-dimensional flow and turbulence properties of each case at the longitudinal transition were plotted with respect to the water depth. Bed shear velocities, $u_*$ were calculated by LOW (Law of the wall) method from the vertical distribution of stream-wise velocity in the ranges of $z^+ (= z u_* / v)$ from 30 to 300 (Kundu and Cohen [18] in Figure 6a). In this study, the outer boundary thickness of logarithmic velocity distribution, $300\, v/u_*$, was assumed as the expecting length scale of largest vortex size where turbulent flow intensely affects the sediment particle movement from the channel bottom. In Figure 6b, vertical distributions of turbulent kinetic energy were plotted with calculated bed shear velocity. For channels with smooth and rough beds, Graf [19] suggested the vertical distribution of turbulent kinetic energy with exponential equation, $k(z) = 3.3 \exp(-2z/h_0)$. The results of this study were in a good agreement of Graf's approach. In Table 2, depth-integrated relative turbulence intensity ($r_0$), which is known as one of the most important factors on the local scour, was calculated. And the particle Reynolds number, $\text{Re}_* (= u_* d_{50} / v)$, sediment factors of experimental cases are presented in the Table 2 also. Results of upstream scour slope were estimated and ranged 2.55–3.34.

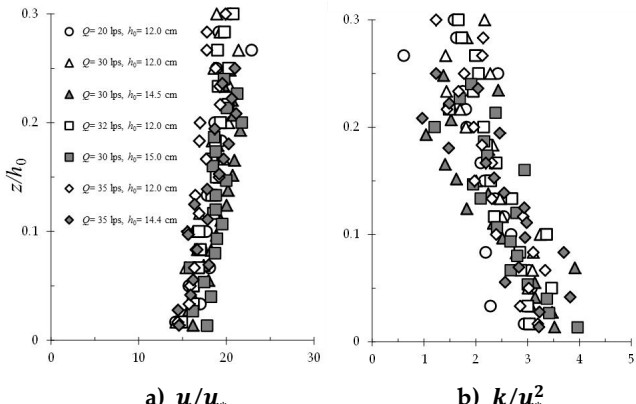

**Figure 6.** Vertical distributions of flow and turbulence at the longitudinal transition. $u$ and $u_*$ denote point velocity and bed shear velocities, respectively. And $k$ is turbulent kinetic energy per unit mass.

**Table 2.** Results of scour test.

| Case Number | $r_0$ (-) | $\cot\theta_u$ (-) | $u_*$ (m/s) | $\tau_0$ (Pa) | Re$_*$(-) | $T$ (hr.) | $y_{m,e}$ (Observed, m) |
|---|---|---|---|---|---|---|---|
| Q20h120d12 | 0.073 | 2.55 | 0.015 | 0.29 | 18.0 | 369 | 0.054 |
| Q30h150d12 | 0.054 | 3.03 | 0.018 | 0.23 | 20.4 | 314 | 0.098 |
| Q30h120d12 | 0.055 | 3.12 | 0.021 | 0.44 | 25.2 | 434 | 0.188 |
| Q35h120d12 | 0.054 | 3.34 | 0.026 | 0.68 | 31.2 | 154 | 0.181 |
| Q35h144d12 | 0.056 | 3.21 | 0.022 | 0.48 | 26.4 | 509 | 0.193 |
| Q30h145d12 | 0.059 | 3.16 | 0.018 | 0.32 | 21.6 | 717 | 0.154 |
| Q20h120d06 | 0.073 | 3.07 | 0.015 | 0.23 | 9.0 | 700 | 0.072 |
| Q35h120d06 | 0.054 | 2.92 | 0.026 | 0.68 | 15.6 | 120 | 0.200 |
| Q35h150d06 | 0.057 | 2.89 | 0.020 | 0.40 | 12.0 | 358 | 0.173 |
| Q32h120d06 | 0.057 | 3.08 | 0.022 | 0.48 | 13.2 | 289 | 0.198 |

## 4.2. Development of Scour Hole and Maximum Scour Depth

The temporal and spatial changes of local scour of all 10 test cases were plotted in Figure 7. As previously mentioned, the upstream and downstream scour slope were approaching to constant values with time. In other words, the scour hole grows and reaches to equilibrium state. The difference of up and downstream scour slope is that the former became constant from the beginning of scour process while the latter slowly approach to a constant value. In the region of downstream scour slope undulation of bed profiles occurred after a certain period time. Values of $y_m$ and $L_1$ gradually increased. Deposition ridges in the downstream of $x = L_s$ were propagated to the downstream with respect to time. In several cases in which approaching velocities are relatively slow, irregularities of bed profile were observed after reaching the stabilized-maximum scour depth. From the temporal changes of bed profiles, values of $y_m(t)$, which were normalized by $\lambda$ and $t_\lambda$ were plotted in Figure 8. Previous equation ($\lambda = h_0$) by Breusers [2] and a new variable, ($\lambda = 300\, v/u_*$) in this study were plotted in Figure 8a,b respectively.

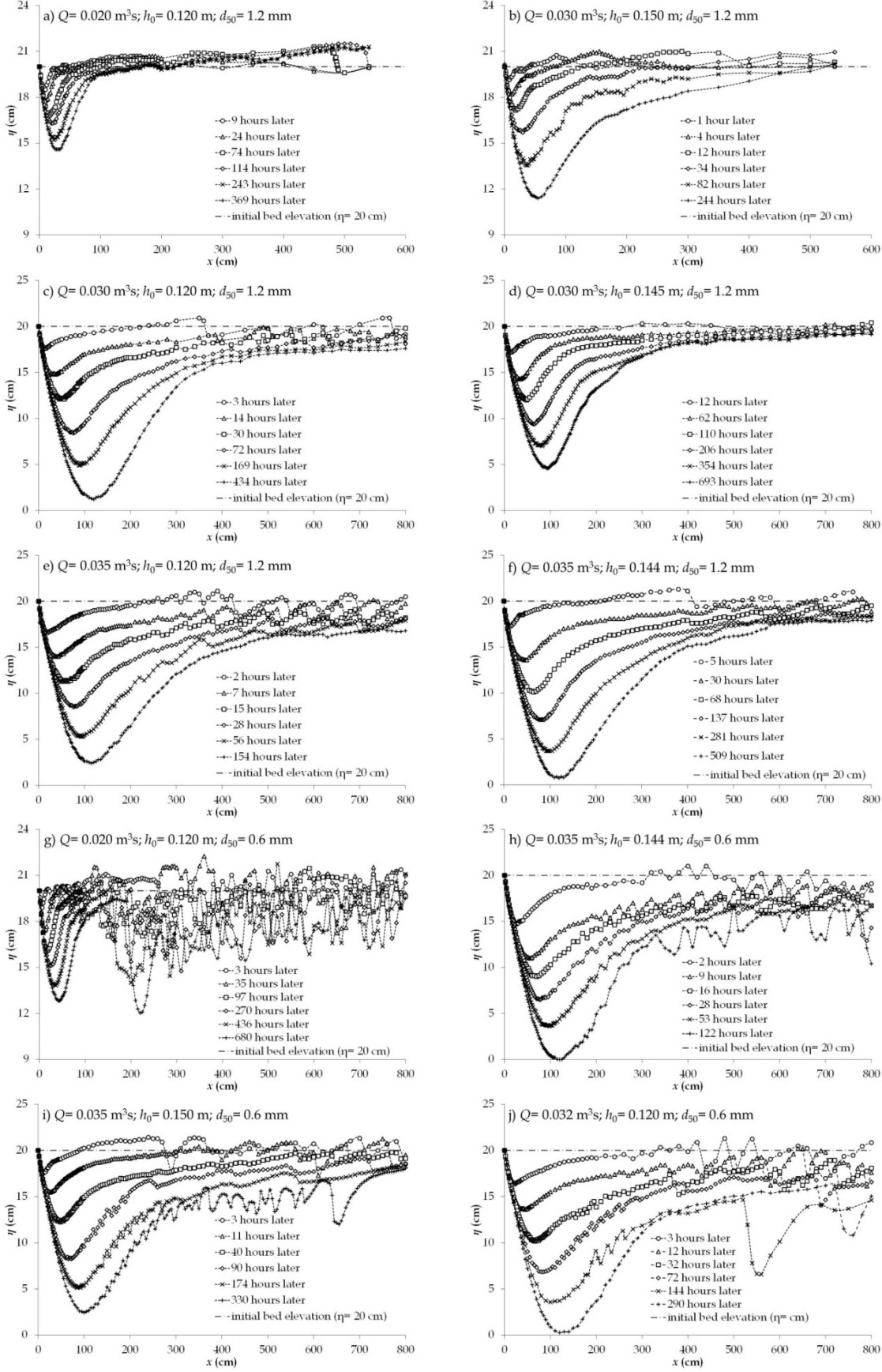

**Figure 7.** Temporal development of scoured bed profiles.

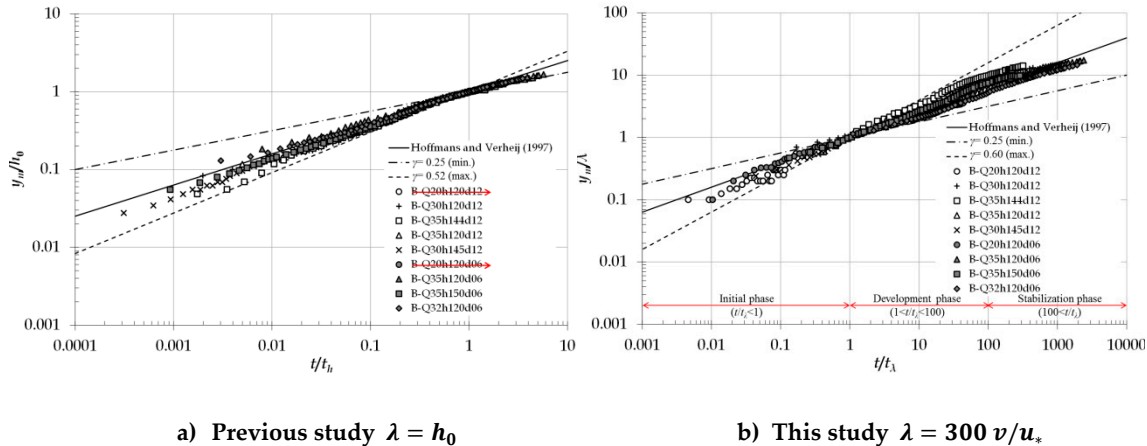

a) **Previous study** $\lambda = h_0$         b) **This study** $\lambda = 300 \, v/u_*$

**Figure 8.** Normalized maximum scour depth as function of time.

Both results of $y_m(t)$ presented that they are increasing exponentially and reached to the equilibrium values of them after a certain time. However, results by Breusers' approach were difficult to be classified into the four-time phase clearly and were not applicable to the case ($y_{m,e} \ll h_0$). Whereas the time phase was easily classified by the method of this study (initial phase: $t/t_\lambda > 1$; development phase: $1 < t/t_\lambda < 100$–1000; stabilization phase: $t/t_\lambda > 100$–1000). Also values of calibration exponent, $\gamma$ of Breusers' approach is estimated in the range from 0.25 to 0.52 and those of this study are estimated from 0.25 to 0.60 (Table 3). Estimated values $u_{*,c}$ and $U_c$ based on the Shields diagram, were varied in the range from 0.016 to 0.023 and from 0.24 to 0.30, respectively. Otherwise, calculated values of $U_c$ based on the Equation (2) were quite different from the measured values in Table 2. Therefore, new approach for the equilibrium values of $y_m$ should be applied. Comparing results of case Q20h120 and case Q35h120, scouring parameter, it was revealed that $\gamma$ of finer bed materials is smaller than coarser bed materials.

**Table 3.** Measured quantities of scour test.

| Case Number | $\lambda$ (m) | $t_\lambda$ (Hour) | $\gamma$ (-) | $u_{*,c}$ (m/s) | $U_c$ (m/s) | $y_{m,e}$ (Calculated, m) |
|---|---|---|---|---|---|---|
| Q20h120d12 | 0.020 | 18.0 | 0.439 | 0.021 | 0.26 | 0.12 |
| Q30h150d12 | 0.017 | 3.80 | 0.400 | 0.023 | 0.30 | 0.18 |
| Q30h120d12 | 0.014 | 0.98 | 0.425 | 0.022 | 0.28 | 0.31 |
| Q35h120d12 | 0.012 | 0.10 | 0.458 | 0.019 | 0.24 | 0.51 |
| Q35h144d12 | 0.014 | 1.60 | 0.463 | 0.023 | 0.30 | 0.29 |
| Q30h145d12 | 0.016 | 3.80 | 0.408 | 0.023 | 0.30 | 0.19 |
| Q20h120d06 | 0.020 | 16.0 | 0.366 | 0.018 | 0.26 | 0.11 |
| Q35h120d06 | 0.012 | 0.05 | 0.356 | 0.016 | 0.23 | 0.55 |
| Q35h150d06 | 0.018 | 1.25 | 0.425 | 0.018 | 0.27 | 0.32 |
| Q32h120d06 | 0.014 | 0.15 | 0.378 | 0.017 | 0.24 | 0.42 |

*4.3. Equilibrium Maximum Scour Depth*

Numerous researches suggested empirical functions to predict the equilibrium scour depth and engineers made an effective design of bed protection downstream of hydraulic structures based on predicted scour depth. As obtained by Termini [10], maximum scour depth of the equilibrium state in this study was determined and compared using, $\partial y_m / \partial t \cong 0 \left( \partial y_m / \partial t = 10^{-6} \right)$.

Applying the Equation (2), values of equilibrium maximum scour depth were estimated in Table 3. In this study based on the previous studies, maximum scour depth of scour hole can be written as follows:

$$\frac{y_{m,e}}{h_0} = \Phi\left(r_0, \frac{\sqrt{gh_0}}{U_0}, \frac{u_* d_{50}}{\nu}\right) \tag{5}$$

And new functional relationship between the previously suggested dominant factor on local scouring, $r_0$, Fr, and Re$_*$ and $y_{m,e}$, which were represented as turbulence properties from the upstream, flow condition, and frictional properties by sediment particle were introduced as follows:

$$\frac{y_{m,e}}{h_0} = r_0{}^{\alpha_1} \mathrm{Fr}^{\alpha_2} \mathrm{Re}_*{}^{\alpha_3} \tag{6}$$

A nonlinear regression method was used to get the parameters of Equation (6). Froude number has the greatest influence on equilibrium state of local scouring ($\alpha_1 = -0.91$, $\alpha_2 = 1.62$, $\alpha_3 = -0.11$) and these results are quite contrary to the results of Gaudio and Marion [16]. In other words, in the cases which Froude number is relatively small, unlike studies of Gaudio and Marion [16], Froude number is most dominant factor on the equilibrium maximum scour depth (Figure 9). With application of Equation (2) by Dietz [3] were not fit on the measured values of this tests (Figure 9a) and we compared the applications of Equation (3) by Gaudio et al. [15] and this study (Figure 9b). This new functional relationship about equilibrium maximum scour depth reflects the flow and turbulence properties with sediment characteristics. From the experimental results, temporal development of bed elevation was revealed and can be used in prediction of the equilibrium state of $y_m$.

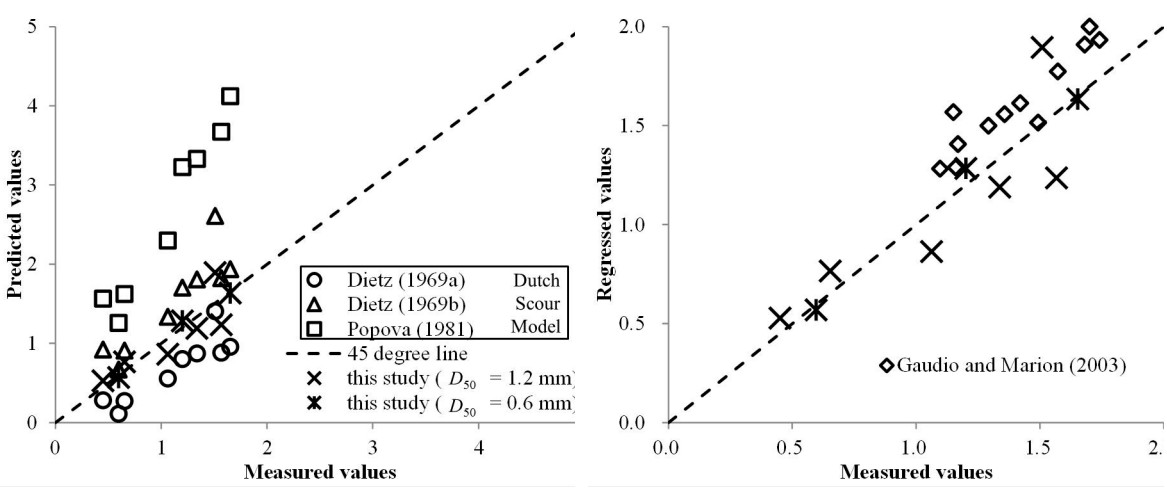

a)   Application of previous studies          b)   Comparison of previous studies

**Figure 9.** Comparison of dimensionless equilibrium maximum scour depth.

## 5. Conclusions and Remarks

In this study, spatial and temporal development of local scouring at the downstream of fixed bed with cases of low Froude numbers were conducted and analyzed with experimental approaches. From the results, it was concluded as follows:

- Laboratory experiments on local scouring at the downstream of fixed bed model were carried out with relatively lower Froude number (less than 0.5) than those of the previous study by Gaudio and Marion [16] and therefore, the total duration of the experiments have exceeded 700 hours. As previous researches have suggested, dominant shapes of local scouring at the downstream of the fixed bed were characterized with time. The upstream scour slope at the downstream of fixed bed reached its equilibrium value earlier than other dimensions and the developments

of the scoured hole were gradually increased with time and reached to the equilibrium state. At downstream of the scoured hole, eroded sediment particles were deposited and propagated toward downstream.

- Temporal development of maximum scour depth, which should have dominantly been considered for the design of the various hydraulic structures in rivers and streams, was analyzed with more reasonable dimensionless time and length scales and classified into the four-time step of the local scour development with revision of previous studies. And the relationship between dimensionless time and length scale of the maximum scour depth in development phase was revealed as the exponential function with exponent equals approximately 0.4. Also, with this modified relationship is applicable to both deep and shallow water depth cases.

- Values of scouring rates of finer sediment cases were larger than those of coarser cases. Therefore, design and construction of hydraulic structures in the rivers and streams near the finer bed materials should be considered with excessive local scouring in short time. Additional experimental researches about various Froude number and sediment size will be compared with this research and empirical equation can be modified.

**Author Contributions:** S.W.P. performed the experimental research, analyzed the data, and wrote the draft version of this manuscript; J.H.H. analyzed the data and edited the manuscript; J.A. revised the manuscript and supervised the project.

**Funding:** This research received funding from National Research Foundation of Korea (NRF) grant funded by Korean government Ministry of Science, ICT & Future Planning 2016R1C1B1014280.

**Acknowledgments:** This work was supported by National Research Foundation of Korea (NRF) grant funded by Korean government Ministry of Science, ICT & Future Planning 2016R1C1B1014280.

**Conflicts of Interest:** The authors declare no conflict of interest.

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
