# Peer review of "Physical Modeling of Spatial and Temporal Development of Local Scour at the Downstream of Bed Protection for Low Froude Number"

_water, doi:10.3390/w11051041_

Round 1
Reviewer 1 Report
This paper elucidated the process and the results of development of equation for predicting scouring depth in natural streams. The most remarkable originality of this paper was developing equations to predict scouring depth in low Froude number condition within long-term duration of the channel flow that was not studied yet by numerous researchers.
In the the reviewer's opinion, this paper is publishable because the proposed scouring equation is based on the robust dataset (Figure 7 and Figure 8 is the good evidence). I recommend publishing the manuscript of this paper.
Author Response
A: Thank for your comments. We extensively revised this paper from the top to improve quality.
Reviewer 2 Report
In this study relatively long-term laboratory experiments of local scouring at the downstream of fixed bed in open channel were conducted with mono-granular sediment bed and analyzed about maximum scour depth and its temporal development.
What is the innovation and the implementation of this study? Also, the conclusions are very weak and must be improved.
Important points:
1. There are a lot of grammar and syntax mistakes. More careful proof-reading should be made to correct many typographical errors in the present manuscript. Extensive editing of English language and style is required.
2. Please remove the photographs from the introduction. The introduction must have only the previous research and studies which are closely to the present work. Please create an additional chapter between introduction and materials-methods and explain the use of the photos.
3. Please replace all the sentences that use first plural, for example not “we modified” but “was modified”.
Author Response
A: Thank for your comments. We extensively revised this paper from the top to improve quality.
Important points:
Q: 1. There are a lot of grammar and syntax mistakes. More careful proof-reading should be made to correct many typographical errors in the present manuscript. Extensive editing of English language and style is required.
A: I corrected as you commented.
Q: 2. Please remove the photographs from the introduction. The introduction must have only the previous research and studies which are closely to the present work. Please create an additional chapter between introduction and materials-methods and explain the use of the photos.
A: I corrected as you commented.
Q: 3. Please replace all the sentences that use first plural, for example not “we modified” but “was modified”.
A: I corrected as you commented.
Reviewer 3 Report
It is necessary to explain better the procedures used and the results obtained especially when in contrast to the previous studies.
According to me any experimental research has to be highly taken into account due to the large amount of work performed. But please explain better the difference between your work and the previous ones.
Author Response

(The authors gave the same response as above.)

Reviewer 4 Report
In the study, the Authors investigated temporal and spatial development of scour below bed protection for low Froude numbers. Based on experimental data, the dimensional analysis of the variables that influence the development of the scour hole has been carried out in order to obtain a semi-empirical formula predicting the maximum scour depth of clear-water at the equilibrium state. The results obtained from this study showed that Froude number was a dominant factor in the prediction of equilibrium maximum scour depth
The topic is of general interest to a broad readership. The study is well presented and contributes to the general understanding of scouring downstream of hydraulic structures. However, from my reading of the paper, it is lacking a fair discussion of the results. I encourage the Authors to provide more comprehensive interpretation of results (e.g. links between scour dimension and the hydraulic characteristics of the flow and sediment size) and to discuss the limitations of the obtained exponential function describing the relationship between dimensionless time and length scale of the maximum scour depth in the development phase.
Please, find below some minor comments
Line 17: Should be ‘analysed for’
Lines 21-24: It is unclear. Please rearrange it.
Line 44: Should be ‘deposited’
Line 87: missing explanation: k0 - depth-averaged turbulent kinetic energy.
Line 91: missing explanation: d50 - grain size for which 50% of the total weight of the sediment is finer
Line 95: missing explanation: g - gravitational acceleration
Lines 103-105: Please change into: ‘And they revealed that the length of the scour hole may be the dominant scale of the whole process until the equilibrium condition was reached, which is controversial’ if these keeps your meaning.
Please avoid starting sentences with ‘and’ (Please revise the manuscript accordingly).
Line 120: Please remove ‘the’ before ‘sand’.
Line 138: There is a typo, should be ‘tap water’.
Line 140: Should be ‘sediment load’
Line 152: missing bracket and comma before Re
Line 158: Please explain why two grain sizes were used in the experiments? Could you please justify the effect of grain size on the results.
Lines 188-189: Please remove ‘were’ before ‘gradually increased’
Line 190: Should be ‘in which’
Lines 214-223: It would be good to link this part with line 87.
Lines 222-223: Please remove ‘from time-averaged one’. It is redundant. The acquisition time of instantenous flow velocities should be clarified. It is unclear whether the profile of turbulent kinetic energy was measured after the scour hole reached its equilibrium state.
Line 241: Should be ‘these results’.
Lines 250-276: Conclusion section sounds more like a summary. Please rearrange it.
Lines 271-276: It is a repetition and should be delated.
More detailed discussion of tables 1 and 2 is advisable, e.g. the relation between shear stress, shear velocity and maximum scour depth for different grain diameters.
Author Response
An answer file is uploaded.
thank you.

Round 2
Reviewer 2 Report
I propose the acceptance of this manuscript.
Author Response
Thank you for your comments and recommendations.